# HMGB1-BoxA gene therapy in reversing cisplatin resistance in non-small cell lung cancer

**Nattapong Puthdee[1], Van-Hieu Mai[2,3], Sirapat Settayanon[4,5], Pithi Chanvorachote[6], Chatchawit Aporntewan[7], Apiwat Mutirangura[5], Chanida Vinayanuwattikun[8]\***

1 C2F-Postdoctoral Fellowship, Chulalongkorn University, Bangkok, Thailand, 2 Faculty of Medicine, University of Health Sciences, Vietnam National University Ho Chi Minh City, Vietnam, 3 Graduate Program in Clinical Sciences, Faculty of Medicine, Chulalongkorn University, Bangkok, Thailand, 4 Interdisciplinary Program of Biomedical Sciences, Graduate School, Chulalongkorn University, Bangkok, Thailand, 5 Center of Excellence in Molecular Genetics of Cancer and Human Diseases, Department of Anatomy, Faculty of Medicine, Chulalongkorn University, Bangkok, Thailand, 6 Cell-Based Drug and Health Product Development Research Unit and Department of Pharmacology and Physiology, Faculty of Pharmaceutical Sciences, Chulalongkorn University, Bangkok, Thailand, 7 Department of Mathematics and Computer Sciences & Omics Sciences and Bioinformatics Center, Faculty of Science, Chulalongkorn University, Bangkok, Thailand, 8 Division of Medical Oncology, Department of Medicine, Faculty of Medicine, Chulalongkorn University and The King Chulalongkorn Memorial Hospital, Bangkok, Thailand

\* Chanida.vi@chula.ac.th

## Abstract

Cisplatin is a widely used chemotherapy agent in the treatment of non-small cell lung cancer (NSCLC). However, its clinical efficacy is limited by the development of drug resistance in patients with NSCLC. Recently, we demonstrated that HMGB1-BoxA gene therapy (BoxA) is an ideal cancer treatment that revitalizes normal cells while promoting cancer cells' DNA break cascade. Increasing cytoplasmic HMGB1 is a reason for cisplatin resistance (Cis-R); in this study, we investigated the potential of BoxA to reverse acquired cisplatin resistance in two NSCLC cell lines, A549/Cis-R, and H460/Cis-R. The growth capacity of these cancer cells was significantly impaired upon BoxA treatment, resulting in a notable reduction in cell viability, colony formation, cancer stemness, and self-renewal capacity. Significantly, BoxA enhanced cisplatin sensitivity and promoted apoptosis in cisplatin-treated Cis-R cells. Furthermore, BoxA altered the subcellular localization of the HMGB1 protein, decreasing its cytoplasmic localization. BoxA can potentially reverse cisplatin resistance by altering the translocation of HMGB1.

## Introduction

Non-small cell lung cancer (NSCLC) is a malignant epithelial neoplasm that counts 80–85% of the lung cancer cases [1,2]. Despite advances in precision medicine, platinum-based chemotherapy is still an upfront or subsequent standard treatment in advanced disease [3]. The development of cisplatin resistance is a common issue that prohibits long-term disease control and leads to poor survival [4,5]. Thus,

**Data availability statement:** All relevant data are within the manuscript and its Supporting information files.

**Funding:** This research was supported by the Chulalongkorn University's Graduate Scholarship Program for ASEAN or Non-ASEAN Countries to HMV, the Donate Fund Program, Faculty of Medicine, Chulalongkorn University (Grant number RA(DO)011/67 to CV, and the Second Century Fund (C2F), Chulalongkorn University to NP and CV.

**Competing interests:** The authors have declared that no competing interests exist.

overcoming cisplatin resistance and maintaining therapeutic responses is a critical challenge in managing NSCLC. Multifactorial changes in several cellular signaling pathways mediate the acquisition of drug resistance [6].

High-mobility group box 1 (HMGB1) contributes to cisplatin resistance. HMGB1 expressions in NSCLC-induced drug-resistant depend on cellular localization [7]. Zhang H et al. demonstrated that HMGB1 was gradually located in cytoplasm while inducing cisplatin resistance A549 cells. Moreover, the HMGB1 translocation desensitizes the effects of other chemotherapeutic drugs, including 5-FU and Oxaliplatin [8]. The cytoplasmic HMGB1 mediates cell autophagy is a proposed mechanism that contributes to cisplatin resistance by inhibiting apoptosis [9–11]. In addition to cisplatin, targeting HMGB1 has been proposed to overcome multidrug resistance problems [12].

HMGB1 BoxA gene therapy, a treatment technology using BoxA of HMGB1 (BoxA) expression plasmid transfection, is an ideal cancer therapeutic technique that promotes normal cell DNA stability but induces a γH2AX-associated DNA break signaling cascade [13]. The HMGB1 protein contains two DNA-binding domains, BoxA and BoxB, and a C-terminal acidic tail [14]. BoxA produces DNA-strengthening epigenetic marks in normal cells called youth-DNA-gap [15]. The youth-DNA-gap comprises BoxA, which plays the role of molecular scissors to produce DNA gaps, SIRT1 deacetylated Histone, and AGO4 methylated DNA [15–17]. The youth-DNA-gap prevents DNA damage by relieving the torsion force of the active DNA double helix, promoting a cellular senescence cascade. The cells are gradually rejuvenated after the senescence process stops [15]. The youth-DNA-gap producing role of HMGB1 BoxA gene therapy has been reported in several *in vitro* and *in vivo* models for preventing DNA damage, cellular rejuvenation, and curing chronic degenerative diseases [15,18–21]. In lung cancer cells, this may be due to defection in the youth-DNA-gap complex; instead of strengthening DNA, the role as molecular scissors of BoxA causes γH2AX-associated DNA breaks as well as the DNA damage cascades, poor cell viability, and proliferation and promotes apoptosis [13].

At outside cells, BoxA peptide can be used as a remedy that acts as a functional antagonist of HMGB1. The disrupting HMGB1 mechanism includes Competitive Receptor Antagonism [22], Redox-Dependent Inactivation [23], and Disruption of Heterocomplex Formation [24]. Because HMGB1 plays a role in cisplatin resistance, we evaluated if HMGB1 BoxA gene therapy modifies cisplatin resistance status by interfering with HMGB1 intracellular distribution similar to the action extracellularly.

## Materials and methods

### Cells culture

Human non-small cell lung cancers A549 (genomic alterations of genes such as *KRAS*, *STK11/KEAP1*, and wild-type *TP53*) and H460 cells (genomic alterations of *KRAS*, *PIK3CA,* and wild-type *TP53*) were cultured in RPMI 1640 medium supplemented with 1% antibiotic-antimycotic, 1% glutamax, 10% FBS and maintained at 37 °C in an atmosphere of 5% $CO_2$. A549 cells (CCL-185™) were obtained from the American Type Culture Collection (ATCC). H460 cells were authenticated by STR profiling from ATCC.

## Generation of cisplatin-resistant cells

Cells were progressively induced to cisplatin resistance by stepwise exposure with gradually increasing doses of cisplatin over five months until reaching a concentration of 1.4 µg/ml (4.67 µM), following a previous report [25]. Cisplatin-resistant (Cis-R) cells, represented by morphology and viability to survive in 2.4-fold cisplatin concentration compared to parental cells, were continuously cultured in a medium containing 1.4 µg/ml cisplatin. The Cis-R model in our study represents acquired cisplatin resistance, mimicking the clinical practice of advanced NSCLC treatment.

## BoxA plasmid transfection

BoxA-GFP and control plasmid DNA were kindly obtained from the Center of Excellence in Molecular Genetics of Human Disease, Department of Anatomy, Faculty of Medicine, Chulalongkorn University. Cells were incubated with medium supplementing 1% FBS for at least 2 hours before transfection. The DNA-lipid complex composed of 2 µg of plasmid DNA was prepared using the lipofectamine 3000 reagents following the manufacturer protocol (Invitrogen, USA). BoxA trans-fected cells for 48 h were used for further experiments.

## Cell viability assay

Cells were plated on the 96-well plate with a density of $8 \times 10^3$ cells/well. On the second day, adherent cells were treated with varying concentrations of cisplatin. After the indicated treatment, cells were incubated with 0.5 mg/ml of MTT (Sigma Aldrich, USA) for 3 hours at 37 ºC and reconstituted by DMSO. The absorbance at 570 nm was measured using Multiskan™ FC microplate photometer (Thermo Scientific, Singapore). The cell viability was calculated using percentages compared to the vector control.

## Clonogenic assay

Cells were seeded at a low density of 500 cells/well in 6-well plates and continuously cultured at 37°C in a 5% $CO_2$ incubator for 10 days. In the cisplatin treatment, cells were treated with 5 µg/ml of cisplatin for 3 hours before dissociation and cell seeding. The colonies were fixed with 4% paraformaldehyde for 15 minutes, followed by staining with 0.1% crystal violet in methanol for 15 minutes. Excess dye was removed by washing with PBS three times and air-drying colonies overnight. The colony numbers were counted. The colony-forming efficacy was calculated by the number of treated cells in the colony x 100/number of (colony) control cells.

## Tumor sphere formation assay

A total of 300 cells/well were resuspended in medium supplemented with 1% FBS. The cell suspension was seeded into a 24-well clear flat bottom, ultra-low attachment multiple-well plate. In the cisplatin treatment, cells were treated with 5 µg/ml of cisplatin for 3 hours. After 10 days of incubation, tumor spheroids were collected and reseeded into 96 well plates, allowing the cells to adhere overnight. The adherent cells were then tested for cell viability by MTT assay.

## Apoptosis assay

Apoptotic cells were detected by staining with PE Annexin V Apoptosis detection kit with 7-AAD (BioLegend, USA, Cat# 640934) and flow cytometry analysis. In cisplatin treatment, cells were treated with 5 µg/ml of cisplatin for 24 hours and were accomplished into a single cell by trypsinization, washed twice with cold PBS, and resuspended in 100 µl of Annexin V binding buffer. The cell suspension was mixed with 2 µL of PE Annexin V and 2 µL of 7-AAD viability staining solution and incubated for 10 minutes. Stained cells were resuspended with 300 µl of Annexin V binding buffer and analyzed by MACSQuant Analyzer 10 Flow Cytometer (Miltenyi Biotec, Germany).

## Western blot

Whole protein lysates were collected using RIPA buffer containing protease and phosphatase inhibitors. The cytoplasmic and nuclear protein extracts were prepared following the previous protocol [15]. Protein concentration was measured using protein assay (Bio-Rad, USA). A total of 20 ug of protein was separated with 4–20% Tris-glycine gradient SDS poly-acrylamide gel electrophoresis and transferred to a nitrocellulose membrane. The membrane was soaked in 5% BSA in TBST buffer for blocking the nonspecific binding and subsequently incubated with the following diluted primary antibodies: anti-HMGB1b(1:1000, Abcam, UK, Cat# ab18256, RRID:AB_444360), anti-beta-actin (13E5) (1:2000, Cell Signaling Technology, USA), anti-p53 (7F5) (1:1000, Cell Signaling Technology, USA), and anti-histone H3 (3H1) (1:1000, Cell Signaling Technology, USA) by overnight at 4 °C. Membrane was incubated for 1 hour with the anti-rabbit IgG, HRP-linked antibody (1:2000, Cell Signaling Technology, USA, Cat# 7074, RRID:AB_2099233) and probed with Pierce ECL Western Blotting Substrate (Thermo Scientific, USA). The visualization was conducted by Azure 300 – Chemiluminescent Imaging System (Azure Biosystems, USA). Protein band intensity was measured by using the ImageJ program (RRID:SCR_003070).

## Results

### Phenotypic evaluation of generated acquired cisplatin-resistant (Cis-R) lung cancer cells

The inducing cisplatin resistance in subclones of A549 and H460 cells was conducted by progressively increasing cisplatin concentrations of growing media. The cell lines were able to be maintained in a medium containing 1.4 µg/ml of cisplatin. Both cisplatin resistance cells (Cis-R) (S1 A, B Fig) displayed enlarged polygonal shapes with variations in cell size and larger nuclei compared to their parental cells, which are called cisplatin-sensitive (Cis-S). Forty-eight hours of cisplatin's half-maximal inhibitory concentrations (IC50) were 13.0 ± 1.0 µg/ml and 21.3 ± 1.2 µg/ml for A549/Cis-S and A549/Cis-R, respectively. While IC50s were 10.0 ± 2.0 µg/ml and 24.0 ± 4.0 µg/ml for H460/Cis-S and H460/Cis-R, respectively. This was represented by an AUC ratio of Cis-R to Cis-S, which indicated 1.6–2.4 folds of cisplatin resistance (S1 C, D Fig).

### Growth inhibitory effect of BoxA on both cisplatin-sensitive (Cis-S) and resistant lung cancer cells (Cis-R)

To evaluate the effect of BoxA on growth capacity. We determined cell viability after BoxA transfection at 24, 48 h, and 72 hours. The results showed that approximately 20% of cell reduction was found in BoxA transfected cells compared to transfected empty vectors (control cells) in all the time points in both Cis-S and Cis-R (Fig 1A, B). The colony formation assay also revealed a 30–50% lower number in BoxA transfected cells than control cells. The impact of BoxA was significant in both Cis-R and Cis-S cells (Fig 1C–F). This implies that BoxA itself has an inhibiting growth effect in NSCLC cell lines. Our results are consistent with the previous report [13].

### BoxA reversal cisplatin resistance (Cis-R) phenotype

A significant decrease in the number of viable cells in the BoxA-existing cells exposed to cisplatin, compared to empty vector control cells, was found. The viable cell was approximately 50% in A549/Cis-S, 30% in A549/Cis-S/BoxA, 70% in A549/Cis-R, and 40% in A549/Cis-R/BoxA cells with 30 µg/ml of cisplatin at 24 hours, (Fig 2A). Likewise, the results for H460 cell line were consistent in the same manner (Fig 2B). The colony formation assay revealed that significantly reduced number in BoxA transfect cell lines compared to empty vector control cells, especially cisplatin-resistant conditions (Fig 2C–F). The colony formation inhibitory effect was consistently observed in A549/Cis-R/BoxA (70% inhibition) and H460/Cis-R/BoxA (59% inhibition) cell lines. Furthermore, the impact of BoxA on the enhancement cisplatin effect was also consistent between H460/Cis-S/BoxA and A549/Cis-S/BoxA (S2 Fig). Reversal cisplatin resistance by transfected BoxA does not affect p53 function, confirmed by western blot (S3 Fig). Transfected BoxA in cisplatin-sensitive (Cis-S) parental cell lines (S4 Fig) did not induce significant apoptosis. Progressive significant apoptosis was seen in transfected BoxA in cisplatin-resistant (Cis-R) and combining transfected BoxA with cisplatin treatment (Cis-R/BoxA) (Fig 2G, H).

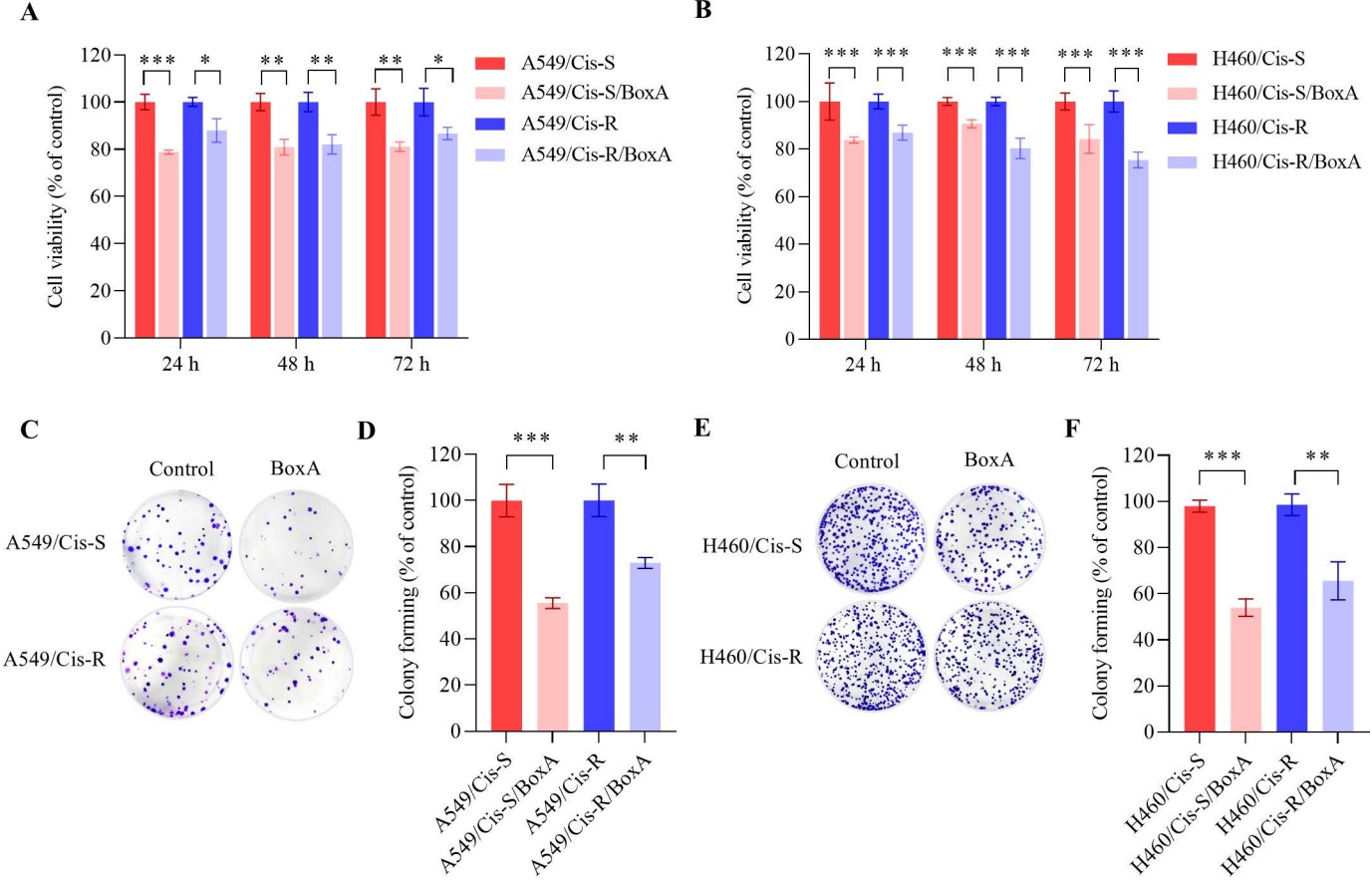

**Fig 1. Growth inhibitory effect of BoxA.** Cell viability of A549/Cis-S, A549/Cis-R cells (A) and H460/Cis-S, H460/Cis-R cells (B), transfected with or without BoxA. The representative colony-forming images and relative colonies-forming efficiency are shown for A549/Cis-S, A549/Cis-R cells with/without BoxA (C, D) and H460/Cis-S, H460/Cis-R cells with/without BoxA (E, F). The results are presented as mean±SD. Statistically significant indicated by *$P<0.05$, **$P<0.01$, and ***$P<0.001$.

Apoptosis enhancement was seen in early and late apoptosis. As a result, BoxA enhanced cisplatin-induced cell apoptosis in acquired cisplatin resistance in wild-type *TP53*.

## BoxA impairs cancer stemness

The Cisplatin-resistance mechanism by induced cancer stem cells (CSCs) subclone in NSCLC has been reported [26,27]. The effects of BoxA on cancer stemness were evaluated by measuring tumor spheroid formation. The size and density of spheroid formation decreased in transfected BoxA A549 and H460 cell lines independent of acquired cisplatin resistance. The effect of decreased CSCs was enhanced with cisplatin-treated conditions (Fig 3A–D and S5 Fig). This suggests that BoxA can reduce cancer stemness, offering a strategy to reverse acquired cisplatin resistance.

## BoxA modulates the subcellular localization of the HMGB1 protein

The overexpression of HMGB1 and its co-localization with P-gp at cytoplasm promoted the malignant progression and the consequence of cisplatin resistance in NSCLC cell lines [7]. We demonstrate diverse effects of cisplatin resistance and transfected BoxA by subcellular localization of HMGB1 expression. Fig 4 represents cytoplasmic and nuclear HMGB1

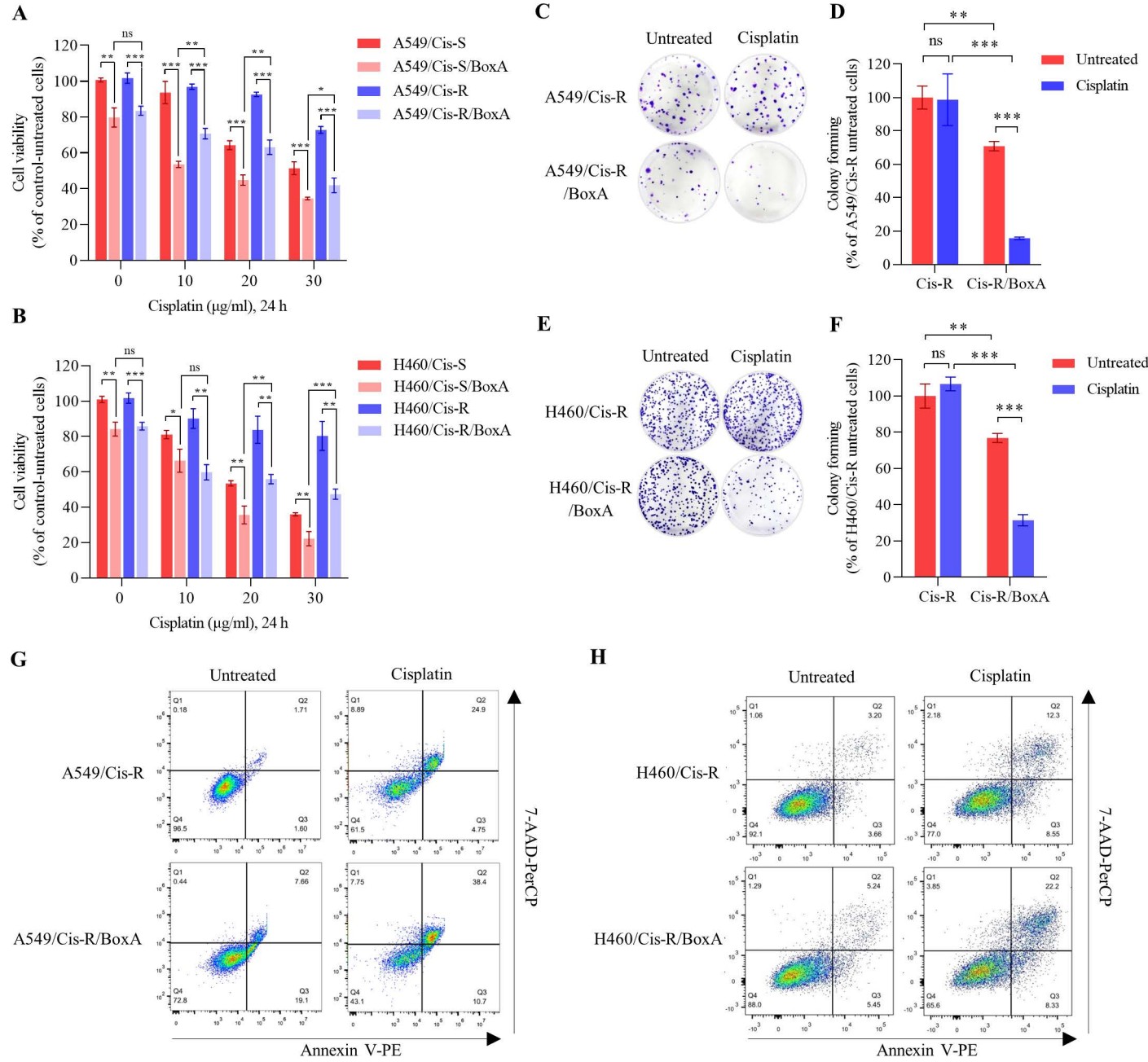

**Fig 2. BoxA improves cisplatin sensitivity of cisplatin -resistant cells.** Cell viability of BoxA-existing- A549/Cis-S, A549/Cis-R cells (A) and H460/Cis-S, H460/Cis-R cells (B) treated with varying concentrations of cisplatin for 24 h. Representative colony-forming results and relative colonies-forming efficiency of A549/Cis-R, A549/Cis-R/BoxA (C, D), and H460/Cis-R, H460/Cis-R/BoxA (E, F) pretreated with 5 µg/ml of cisplatin. Quantitative apoptosis cells after exposure with 30 µg/ml of cisplatin for 24 h, A549/Cis-R, A549/Cis-R/BoxA (G), H460/Cis-R, and H460/Cis-R/BoxA (H). Bar graphs are presented as mean ± SD. The statistically significant is indicated by * $P < 0.05$, ** $P < 0.01$, and *** $P < 0.001$. ns means not significant.

expression in Cis-S, Cis-R/BoxA and Cis-R. There is a slightly increased HMBG1 nuclear expression in H460/Cis-R/BoxA compared to H460/Cis-R. However, this finding is not seen in A549/CisR/BoxA vs. A549/CisR. One possible mechanism of HMGB1 and cisplatin-resistance was related to the subcellular co-localization between HMGB1 and P-gp protein [7,9].

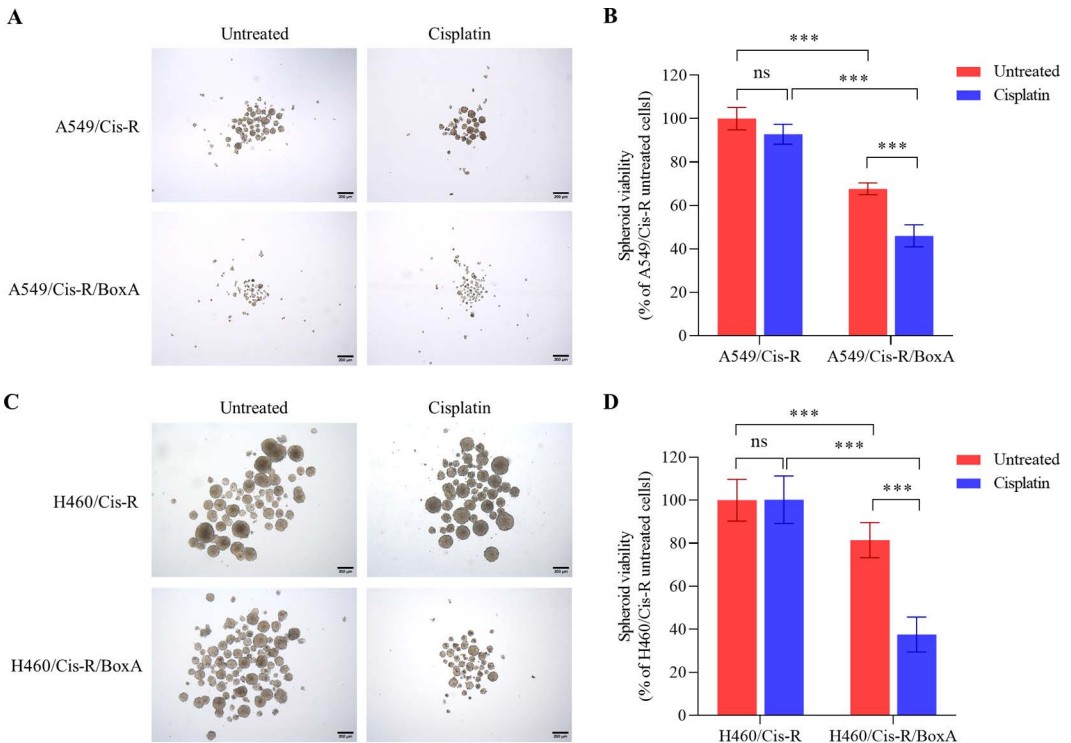

**Fig 3. BoxA impairs the stemness of cisplatin-resistant cells.** Representative tumors spheroids image and relative tumor-spheroids viability of A549/Cis-R, A549/Cis-R/BoxA (A, B) and H460/Cis-R, H460/Cis-R/BoxA (C, D) cell pre-treated with 5 µg/ml of cisplatin. Data are presented as mean±SD. Statistically significant indicated by *** $P < 0.001$. ns means not significant.

Transfected BoxA in cisplatin-resistance (Cis-R/BoxA) significantly reduced the cytoplasmic HMGB1 expression in both A549/Cis-R/BoxA and H460/Cis-R/BoxA compared to Cis-R cell lines. Our study confirmed that HMGB1 cytoplasmic localization expression is modulated by cisplatin resistance and can be reversed by BoxA transfection.

## Discussion

We generated acquired cisplatin resistance of 2 non-small cell lung cancer cell lines (Cis-R) that mimic real clinic situations and tested the outcome of HMGB1 BoxA gene therapy. We found that transfected BoxA plasmid *in vitro* inhibited Cis-S and Cis-R lung cancer growth and increased cisplatin sensitivity in cisplatin resistance models. While CSCs are one mechanism promoting cisplatin resistance, BoxA also reduces Cis-R stemness. Moreover, the acquired cisplatin resistance cell line has more cytoplasmic and less nuclear HMGB1 than Cis-S. The effect of transfected BoxA to reduce cytoplasmic and increased nuclear HMGB1 was the mechanism of lowering cancer aggressiveness of cisplatin resistance NSCLC.

HMGB1 BoxA gene therapy has two molecular consequences in NSCLC cells. Previously, we demonstrated that BoxA reduced NSCLC growth by the molecular scissors action in producing γH2AX-associated DNA brake. Here, we showed that BoxA upregulation reduced cytoplasmic HMGB1 and increased nuclear HMGB1. We observed the reduction of cell growth in both Cis-S and Cis-R. Therefore, molecular scissors mechanism is unlikely to underlie the observed reversal of cisplatin resistance. Since several reports are suggesting cytoplasmic HMGB1 role in cisplatin resistance [7,9] and BoxA gene therapy reverses cisplatin resistance simultaneously with cytoplasmic HMGB1 reduction. This confirms our conclusion that BoxA reversed cisplatin resistance by reducing cytoplasmic HMGB1. Several possible mechanisms explain

…

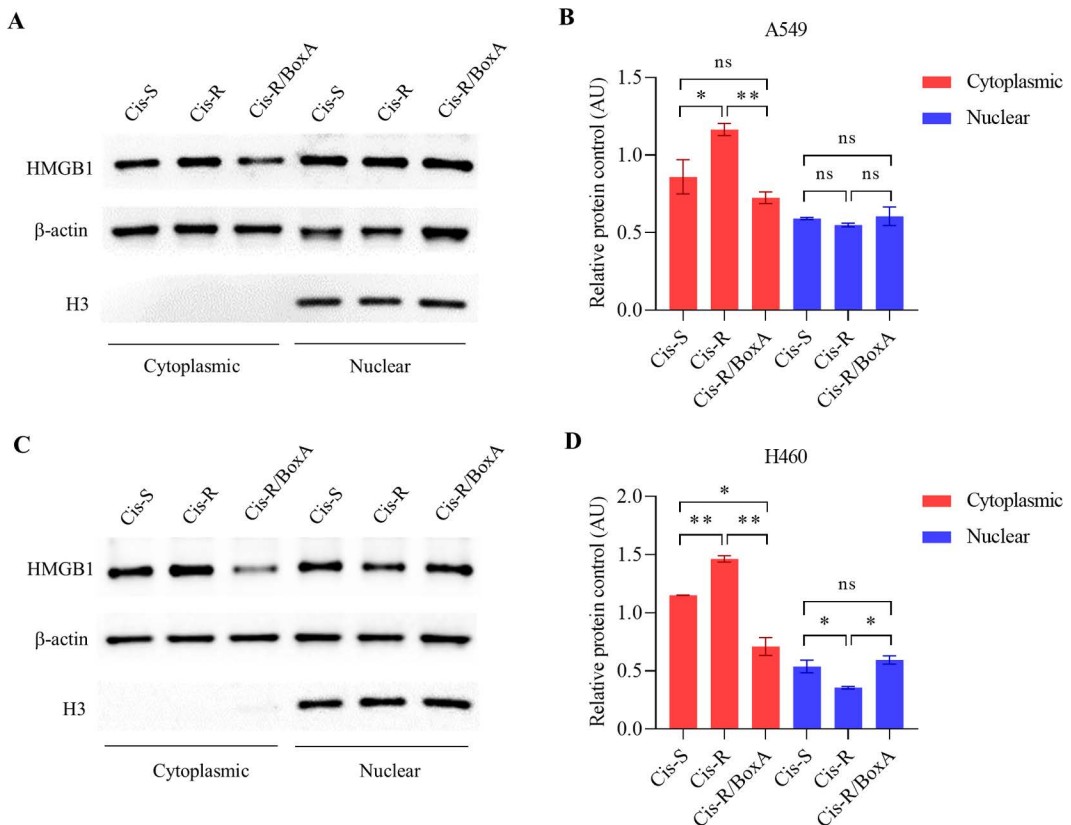

**Fig 4. BoxA modulates subcellular translocation of HMGB1 protein.** HMGB1 protein in subcellular fractionation, cytoplasmic, nuclear are shown by western blot, A549 cells (A), H460 cells (C) and relative HMGB1 expression level for A549 cells (B) and H460 cells (D). Histone H3 is used for nuclear protein -loading control and β-actin is used for cytoplasmic protein- loading control. The statistically significant is indicated by * $P<0.05$, ** $P<0.01$.

how BoxA upregulation can reduce cytoplasmic HMGB1, including decreased production, degradation, or reverse the co-localization of cytoplasmic HMGB1 and P-gp protein in cisplatin resistance. Furthermore, nuclear HMGB1 function in the setting of excessive DNA damage is the enhancement of DNA bending, disrupting DNA repair mechanisms, and facilitating apoptosis [28]. In *vitro*, transfected BoxA of HMGB1 in lung cancer cell lines reduced cell proliferation, promoting apoptosis, and induced the formation of γH2AX-associated DNA double-strand breaks, and DNA damage response (DDR) related protein [13]. Increased DDR-related protein might resume the action of cisplatin on lung cancer [29]. Cisplatin resistance due to cytoplasmic HMGB1 was also reported in other cancers, such as cervical cancer [9], ovarian cancer [10], and leukemia [30]. Also, HMGB1 was proposed as a drug resistance mechanism of different chemotherapeutic agents, including docetaxel, doxorubicin, gemcitabine [31–33]. Future research should be performed to prove if HMGB1 BoxA gene therapy can reverse the drug resistance of those cancers.

Contrary to the effect of BoxA in cancer cells, Transfected boxA itself in non-cancerous cells (HEK293) previously reported the reverse outcomes, contrary to those in cancer cells. It increased proliferation, cell viability, and reduced DNA damage by reducing DDR signaling [13]. This action might stop the DNA damage-senescence-chronic inflammation cascade due to chemotherapeutic agents. We believed that BoxA gene therapy might improve the overall health of cancer patients undergoing chemotherapy treatment. Secondly, the central part of our study is that BoxA-produced DNA gaps increase sensitivity to DNA-damaging agents. Our study did not conduct *in vivo* efficacy and safety studies, which are generally conducted before the first-in-human clinical trial. However, the *in vivo* efficacy and safety of BoxA gene therapy

in other conditions, such as bleomycin-induced pulmonary fibrosis [21] and burn wounds [34], have been reported. We proposed that future clinical trials may be set up using HMGB1 BoxA gene therapy with chemotherapy.

## Conclusions

In conclusion, in addition to strengthening the DNA of normal cells and breaking the DNA of NSCLC cells, HMGB1 gene therapy can reverse the cisplatin resistance of Cis-R NSCLC cells by reducing cytoplasmic HMGB1. Future clinical research is crucial to determine if BoxA is a remedy that can improve the treatment outcome of acquired cisplatin resistance NSCLC patients.

## Supporting information

**S1 Fig. Characterization of cisplatin-resistant cell lines.** Representative image of A549/Cis-R (A), H460/Cis-R (B), and their Cis-S cells. Cisplatin sensitivity of A549/Cis-R and A549/Cis-S cells (C), H460/Cis-R and H460/Cis-S cells (D). Data are presented as mean ± SD of three independent experiments.
(TIF)

**S2 Fig. BoxA augments the growth inhibitory effect of cisplatin in lung cancer cells.** Representative images and colony-forming efficiency of A549/Cis-S/BoxA (A), and H460/Cis-S/BoxA (B) pretreated with 5 µg/ml cisplatin compared to the control-pretreated cells. Bar charts present mean ± SD. Statistically significant is indicated by *** $P < 0.001$.
(TIF)

**S3 Fig. Alterations of p53 protein expression in either context of lung cancer cells.** Protein expression levels of p53 in A549/Cis-S, A549/Cis-R, A549/Cis-R/BoxA, H460/Cis-S, H460/Cis-R, and H460/Cis-R/BoxA cells were evaluated by western blot (A). Relative protein intensity of p53 to protein loading controls β-actin in indicated samples (B).
(TIF)

**S4 Fig. Effect of BoxA on apoptosis induction in lung cancer cells.** A549/Cis-S/BoxA (A), H460/Cis-S/BoxA (B), and their control cells were determined by apoptosis by using Annexin V/7-AAD staining and flow cytometry analysis. Flow cytometry histograms show the percentage rate of necrosis (Q1: 7-AAD+/AnnexinV−), late apoptosis (Q2: 7-AAD+/AnnexinV+), early apoptosis (Q3: 7AAD−/AnnexinV+) and live cells (Q4: 7-AAD−/AnnexinV−).
(TIF)

**S5 Fig. BoxA reduces the stemness of lung cancer cells.** Representative spheroids images and relative spheroid viability of BoxA-existing cells, A549/Cis-S/BoxA (A, B), H460/Cis-S/BoxA (C, D) and their control cells with or without pretreated 5 ug/ml of cisplatin, Bar charts present mean ± SD. Statistically significant is indicated by *** $P < 0.001$.
(TIF)

**S1 S_raw_image. Included the original images, i.e., Western blot, colony formation assay, and sphere formation of the experiment.**
(PDF)

**S1 File. Metadata and raw data of figures and supplementary figures.**
(XLSX)

## Acknowledgments

The authors would like to express our sincere gratitude to the Division of Medical Oncology, Department of Medicine, and the Center of Excellence in Molecular Genetics of Cancer and Human Diseases, Department of Anatomy, Faculty of Medicine, Chulalongkorn University, for providing the necessary research materials and facilities.

## Author contributions

**Conceptualization:** Nattapong Puthdee, Pithi Chanvorachote, Apiwat Mutirangura, Chanida Vinayanuwattikun.

**Data curation:** Nattapong Puthdee.

**Formal analysis:** Nattapong Puthdee, Van-Hieu Mai, Chatchawit Aporntewan.

**Funding acquisition:** Nattapong Puthdee, Van-Hieu Mai, Chanida Vinayanuwattikun.

**Investigation:** Nattapong Puthdee.

**Methodology:** Nattapong Puthdee.

**Project administration:** Apiwat Mutirangura, Chanida Vinayanuwattikun.

**Resources:** Sirapat Settayanon.

**Supervision:** Apiwat Mutirangura, Chanida Vinayanuwattikun.

**Validation:** Nattapong Puthdee.

**Visualization:** Nattapong Puthdee.

**Writing – original draft:** Nattapong Puthdee.

**Writing – review & editing:** Apiwat Mutirangura, Chanida Vinayanuwattikun.

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
