## [Decision Letter · Decision Letter 0]

Dear Dr. Vinayanuwattikun,

Thank you for submitting your manuscript to PLOS ONE. After careful consideration, we feel that it has merit but does not fully meet PLOS ONE’s publication criteria as it currently stands. Therefore, we invite you to submit a revised version of the manuscript that addresses the points raised during the review process.

We look forward to receiving your revised manuscript.

Kind regards,

Vino Cheriyan, PhD

Academic Editor

PLOS ONE

“This research was supported by the Chulalongkorn University’s Graduate Scholarship Program for ASEAN or Non-ASEAN Countries to HMV, the Donate Fund Program, Faculty of Medicine, Chulalongkorn University (Grant number RA(DO)011/67 to CV, and the Second Century Fund (C2F), Chulalongkorn University to NP and CV.”

Additional Editor Comments:

1.5 micrograms/ml of cisplatin insufficient to maintain drug-resistant cells. The cells must be able to survive in 5 micrograms of cisplatin per milliliter if it is resistant. From the viability assay (S Fig.1) the drug-resistant cells appear not so resistant to cisplatin.

How long are drug-resistant cells kept in cisplatin at a concentration of 1.5 micrograms per milliliter?

I would suggest selecting (randomly picking from the drug-resistant cells) a minimum of five clones from the drug-resistant cells and performing viability tests.

I was unable to identify any major differences in the protein expression between drug-resistant cells and HMBG1 control cells. Do the HMBGI expression on clones that are resistant to drugs. Also check potential markers of drug resistance in NSCLC.

Even though this manuscript lacks the in vivo experiments I suggest authors point out some more evidence of the mechanism of cisplatin resistance in these cell lines.

Reviewers' comments:

Reviewer's Responses to Questions

**Comments to the Author**

1. Is the manuscript technically sound, and do the data support the conclusions?

Reviewer #1: Yes

Reviewer #2: Yes

Reviewer #3: Yes

2. Has the statistical analysis been performed appropriately and rigorously?

Reviewer #1: Yes

Reviewer #2: Yes

Reviewer #3: Yes

3. Have the authors made all data underlying the findings in their manuscript fully available?

Reviewer #1: Yes

Reviewer #2: Yes

Reviewer #3: Yes

4. Is the manuscript presented in an intelligible fashion and written in standard English?

Reviewer #1: Yes

Reviewer #2: Yes

Reviewer #3: Yes

Reviewer #1: The current manuscript presents novel approach to the possible treatment of small cell lung cancer utilizing a practical approach to the scientific question at hand. Manuscript is written in a stepwise manner, and methods are elaborately described. However, grammatical errors, formatting inconsistencies and missing references need to be revised throughout the paper. Bar scale needs to be added in Fig.3.

Reviewer #2: Review Comments to the Author

The manuscript titled “HMGB1-BoxA gene therapy in reversing cisplatin resistance in non-small cell lung cancer”presents compelling findings on the modulation of HMGB1 localization and its role in overcoming acquired cisplatin resistance in NSCLC models. The work is timely and adds valuable insights to the understanding of chemotherapy resistance mechanisms. The manuscript is generally well-organized, and the experimental data are sound. However, I recommend the following minor revisions to improve clarity, data interpretation, and completeness.

1. Data Presentation and Figure Clarity

• Lines 169–193 (Figure 2): Please clarify the differential impact of BoxA treatment on Cis-S vs. Cis-R cell lines. Consider adding annotations or group labels to the figure to facilitate direct comparison.

• Lines 203–206 (Figure 3): Include scale bars and indicate the magnification used in the tumor spheroid images to enhance reproducibility and visual interpretation.

2. Discussion – Mechanistic Clarification and Language Precision

• Lines 233–245: Expand the explanation on how nuclear HMGB1 contributes to increased chemotherapy sensitivity. You may consider referencing mechanisms such as chromatin stabilization, transcriptional regulation, or modulation of DNA repair processes.

• Lines 243–244: The suggested mechanism involving ‘heteroduplex formation’ is interesting but speculative. Please explicitly state this as a hypothesis and recommend that future experiments validate this proposed mechanism.

3. BoxA Toxicity in Non-Cancerous Cells

• The manuscript does not include data on the potential off-target or toxic effects of BoxA in non-cancerous cells. This is particularly relevant given its potential translational use. If such data are unavailable, consider including this as a limitation in the Discussion and propose its evaluation in future studies.

• Specifically, suggest whether experiments on normal lung epithelial cells (e.g., BEAS-2B) could help assess selective toxicity.

4. In Vivo Validation and Translational Relevance

• While the current in vitro findings are promising, the manuscript lacks in vivo validation of BoxA’s therapeutic effect. Including (or proposing) in vivo efficacy studies—for instance, using cisplatin-resistant NSCLC xenograft or PDX models—would greatly strengthen the translational relevance of the work. Such studies could assess BoxA’s ability to restore cisplatin sensitivity, evaluate toxicity in normal tissues, and confirm the modulation of HMGB1 localization in a physiologically relevant context.

• If in vivo data are currently unavailable, we recommend discussing this limitation clearly and outlining future directions that involve animal models and pharmacokinetic evaluations of BoxA gene therapy.

5. Language, Grammar, and Abbreviation Consistency

• Line 34 (Abstract): “These BoxA can potentially...” → revise to: “BoxA can potentially...”

• Line 70: Consider replacing “alters cisplatin resistance status” with “modifies cisplatin resistance status” for precision.

• Lines 237–238: The phrase “action of the molecular scissors is unlikely to be the mechanism...” can be reworded for clarity. Suggested revision:

“The molecular scissors mechanism is unlikely to underlie the observed reversal of cisplatin resistance.”

• Ensure consistent use of abbreviations, especially Cis-S (cisplatin-sensitive) and Cis-R (cisplatin-resistant), throughout the text and figures.

Overall Recommendation: Minor Revision

The manuscript offers strong potential for advancing therapeutic strategies in cisplatin-resistant NSCLC. Addressing the points above, particularly the addition of BoxA toxicity profiling, and planning or discussing in vivo validation, will significantly enhance the manuscript’s impact and translational value.

Reviewer #3: I appreciate the authors for the work leading to this manuscript. Based on the experimental evidence and the interpretation of the results, I have few concerns:

HMGB1-BoxA gene therapy is argued to reverse the cisplatin induced chemoresistance. However, there is no data that confirms the regulation of ABC transporters upon acquired resistance and its reversal following therapy. I would suggest, a western blot to confirm this.

Have the authors confirmed the presence of stem cells by either flow cytometry or AldeFluor assay?

Lastly, do the authors plan any in vivo experiments to validate their findings in vitro?

**Do you want your identity to be public for this peer review?** For information about this choice, including consent withdrawal, please see our Privacy Policy

Reviewer #1: No

Reviewer #2: **Yes: ** Venkatesh Mayandi

Reviewer #3: No

---

## [Author Response · Author response to Decision Letter 1]

9 May 2025

Response to Editor

1.5 micrograms/ml of cisplatin insufficient to maintain drug-resistant cells. The cells must be able to survive in 5 micrograms of cisplatin per milliliter if it is resistant. From the viability assay (S Fig.1) the drug-resistant cells appear not so resistant to cisplatin. How long are drug-resistant cells kept in cisplatin at a concentration of 1.5 micrograms per milliliter? I would suggest selecting (randomly picking from the drug-resistant cells) a minimum of five clones from the drug-resistant cells and performing viability tests.

Ans: We acknowledge this concern. Our cisplatin-resistant cells were maintained in 1.4 μg/ml (4.67 μM) cisplatin for over five months before being used in subsequent validation experiments. The protocol for generating cisplatin-resistant cells closely followed a previous report. The resistant cells were assessed through changes in cell morphology, cisplatin inhibition 50 (IC50) using the MTT assay. Similarly, we defined Cis-R based on morphology, and 2.4-fold resistance to cisplatin compared to the parental cells. These results support that our Cis-R cells represent a model of acquired cisplatin resistance. We added information in the method section (line: 82-88).

I was unable to identify any major differences in the protein expression between drug-resistant cells and HMBG1 control cells. Do the HMBGI expression on clones that are resistant to drugs.

Ans: To clarify more, the author demonstrates diverse effects of cisplatin resistance and transfected BoxA by subcellular localization of HMGB1 expression. Figure 4 represents cytoplasmic and nuclear HMGB1 expression in Cis-R/BoxA and Cis-R. There is a slightly increased HMBG1 nuclear expression in H460/Cis-R/BoxA compared to H460/Cis-R. However, this finding is not seen in A549/CisR/BoxA vs. A549/CisR. However, the cytoplasmic HMGB1 expression is a diverse phenomenon. One possible mechanism of HMGB1 and cisplatin-resistance was related to the subcellular co-localization between HMGB1 and P-gp protein. Our study confirmed that HMGB1 cytoplasmic localization expression is modulated by cisplatin resistance and can be reversed by BoxA transfection. We revised to add more information in lines 214-224.

Review Comments to the Author

The manuscript titled “HMGB1-BoxA gene therapy in reversing cisplatin resistance in non-small cell lung cancer”presents compelling findings on the modulation of HMGB1 localization and its role in overcoming acquired cisplatin resistance in NSCLC models. The work is timely and adds valuable insights to the understanding of chemotherapy resistance mechanisms. The manuscript is generally well-organized, and the experimental data are sound. However, I recommend the following minor revisions to improve clarity, data interpretation, and completeness.

1. Data Presentation and Figure Clarity

• Lines 169–193 (Figure 2): Please clarify the differential impact of BoxA treatment on Cis-S vs. Cis-R cell lines. Consider adding annotations or group labels to the figure to facilitate direct comparison.

Ans: Agree. Our reports demonstrated that in vitro, transfected BoxA of HMGB1 in lung cancer cell lines itself reduced cell proliferation, cell migration, and promoted apoptosis, consistent with previous reports. This section is demonstrated in Figure 1. We edited the results by adding information in lines 163-164. Figure 2 shows the results of combining treated cisplatin in infected BoxA in A549 and H460 cell lines. The combined treatment enhanced cisplatin sensitivity, which more markedly reduces cell viability in treated cisplatin of Cis-S and Cis-R cells. Moreover, we have revised the figure by adding annotation lines with asterisks to indicate the statistically significant differences between Cis-S/BoxA and Cis-R/BoxA.

• Lines 203–206 (Figure 3): Include scale bars and indicate the magnification used in the tumor spheroid images to enhance reproducibility and visual interpretation.

Ans: Agree. We added the scale bars with magnification to the tumor spheroid images and updated.

2. Discussion – Mechanistic Clarification and Language Precision

• Lines 233–245: Expand the explanation on how nuclear HMGB1 contributes to increased chemotherapy sensitivity. You may consider referencing mechanisms such as chromatin stabilization, transcriptional regulation, or modulation of DNA repair processes.

• Lines 243–244: The suggested mechanism involving ‘heteroduplex formation’ is interesting but speculative. Please explicitly state this as a hypothesis and recommend that future experiments validate this proposed mechanism.

Ans: Agree. We regret the overstatement and appreciate the suggestion. The role of nuclear HMGB1 function in the setting of excessive DNA damage is the enhancement of DNA bending, disrupting DNA repair mechanisms, and facilitating apoptosis. In vitro, transfected BoxA of HMGB1 in lung cancer cell lines itself reduced cell proliferation, promoting apoptosis, and induced the formation of γH2AX-associated DNA double-strand breaks, and DNA damage response (DDR) related protein. Increased DDR-related protein might resume the action of cisplatin on lung cancer. Another mechanism is the reversal of the co-localization of cytoplasmic HMGB1 and P-gp protein in cisplatin resistance. These possibly support the idea that BoxA enhances cellular sensitivity to chemotherapy. We revised the discussion section in lines 250–256.

3. BoxA Toxicity in Non-Cancerous Cells

• The manuscript does not include data on the potential off-target or toxic effects of BoxA in non-cancerous cells. This is particularly relevant given its potential translational use. If such data are unavailable, consider including this as a limitation in the Discussion and propose its evaluation in future studies.

• Specifically, suggest whether experiments on normal lung epithelial cells (e.g., BEAS-2B) could help assess selective toxicity.

Ans: We agree on the opinion of showing the effect of BoxA on non-cancerous cells. We revised the previous discussion to be clearer. Transfected boxA in non-cancerous cells (HEK293) previously reported the reverse outcomes, compared to cancer cells: increased proliferation, cell viability, and reduced DNA damage by reduced DDR signaling. We added this information in lines 261-265.

4. In Vivo Validation and Translational Relevance

• While the current in vitro findings are promising, the manuscript lacks in vivo validation of BoxA’s therapeutic effect. Including (or proposing) in vivo efficacy studies—for instance, using cisplatin-resistant NSCLC xenograft or PDX models—would greatly strengthen the translational relevance of the work. Such studies could assess BoxA’s ability to restore cisplatin sensitivity, evaluate toxicity in normal tissues, and confirm the modulation of HMGB1 localization in a physiologically relevant context.

• If in vivo data are currently unavailable, we recommend discussing this limitation clearly and outlining future directions that involve animal models and pharmacokinetic evaluations of BoxA gene therapy.

Ans: The author agrees with the reviewer that in vivo efficacy studies are generally conducted before a phase I, first-in-human clinical trial. However, we regret the limitation of performing this experiment due to time consumption and budget constraints. Moreover, the vivo efficacy of Box A gene therapy in other conditions has been reported, such as bleomycin-induced pulmonary fibrosis and burn wound treatment. We added this limitation and placed more information on in vivo efficacy in the discussion (line 267-272).

5. Language, Grammar, and Abbreviation Consistency

• Line 34 (Abstract): “These BoxA can potentially...” → revise to: “BoxA can potentially...”

• Line 70: Consider replacing “alters cisplatin resistance status” with “modifies cisplatin resistance status” for precision.

• Lines 237–238: The phrase “action of the molecular scissors is unlikely to be the mechanism...” can be reworded for clarity. Suggested revision:

“The molecular scissors mechanism is unlikely to underlie the observed reversal of cisplatin resistance.”

• Ensure consistent use of abbreviations, especially Cis-S (cisplatin-sensitive) and Cis-R (cisplatin-resistant), throughout the text and figures.

Ans: The author would like to thank reviewers for revising some grammatical errors to make it more meaningful. We revised in line 34, 70, 244-245. Moreover, we used Cis-S and Cis-R to represent cisplatin-sensitive and cisplatin-resistant throughout the text, except for the main title or sub-title, where we used both the abbreviation and the full description.

---

## [Decision Letter · Decision Letter 1]

HMGB1-BoxA gene therapy in reversing cisplatin resistance in non-small cell lung cancer

PONE-D-25-14742R1

Dear Dr. Vinayanuwattikun,

We’re pleased to inform you that your manuscript has been judged scientifically suitable for publication and will be formally accepted for publication once it meets all outstanding technical requirements.

Kind regards,

Vino Cheriyan, PhD

Academic Editor

PLOS ONE

The revised manuscript is of good quality with well-organized experimental design, methods are explained in detail. and a novel approach. In my opinion, this manuscript is both scientifically sound and interesting for the scientific community and PLOS ONE readers.

The authors have adequately addressed all the comments addressed by the reviewers and I recommend the manuscript for the publication in PLOS ONE. I appreciate the authors for the sincere attempt to revise the manuscript.

Reviewers' comments:

Reviewer's Responses to Questions

**Comments to the Author**

Reviewer #1: All comments have been addressed

Reviewer #2: All comments have been addressed

2. Is the manuscript technically sound, and do the data support the conclusions?

Reviewer #1: Yes

Reviewer #2: Yes

3. Has the statistical analysis been performed appropriately and rigorously?

Reviewer #1: Yes

Reviewer #2: Yes

4. Have the authors made all data underlying the findings in their manuscript fully available?

Reviewer #1: Yes

Reviewer #2: Yes

5. Is the manuscript presented in an intelligible fashion and written in standard English?

Reviewer #1: Yes

Reviewer #2: Yes

Reviewer #1: All the comments presented earlier for the original submission have been addressed; authors have made a sincere attempt for revision.

Reviewer #2: Review Comments to the Author: The revised manuscript presents a timely and significant contribution to the field of cancer therapeutics by elucidating the role of HMGB1-BoxA gene therapy in overcoming cisplatin resistance in NSCLC. The authors have adequately addressed prior concerns, and the scientific rigor of the study is evident in the well-structured experimental design, comprehensive methodology, and mechanistic insights.

Strengths:

(1) The study is well-motivated, addressing a critical challenge in chemotherapy resistance.

(2) The authors provide robust in vitro data demonstrating the efficacy of BoxA in reducing cell viability, stemness, and reversing cisplatin resistance.

(3) The addition of mechanistic details, especially the modulation of subcellular localization of HMGB1, significantly strengthens the biological interpretation.

Conclusion: This study provides a novel and compelling approach to tackling drug resistance in NSCLC and is of interest to the cancer research and translational therapeutics community. I recommend the manuscript for publication in PLOS ONE following final editorial checks.

**Do you want your identity to be public for this peer review?** For information about this choice, including consent withdrawal, please see our Privacy Policy

Reviewer #1: No

Reviewer #2: **Yes: ** Venkatesh Mayandi

---

## [Editor Report · Acceptance letter]

PONE-D-25-14742R1

PLOS ONE

Dear Dr. Vinayanuwattikun,

I'm pleased to inform you that your manuscript has been deemed suitable for publication in PLOS ONE. Congratulations! Your manuscript is now being handed over to our production team.

Kind regards,

on behalf of

Dr. Vino Cheriyan

Academic Editor

PLOS ONE